# Awareness of diagnosis, treatment plan and prognosis among patients attending public hospitals and health centers in Addis Ababa, Ethiopia

**Alula M. Teklu**[1]*, **Mebratu Abraha**[2], **Tegenne Legesse**[3], **Mahteme Bekele**[2], **Abrham Getachew**[2], **Bizuayehu Aseffa**[4], **Million Molla**[4], **Frehiwot Belachew**[1], **Tilahun N. Haregu**[5]

**1** MERQ consultancy PLC, Addis Ababa, Ethiopia, **2** Saint Paul Hospital Millennium Medical College, Addis Ababa, Ethiopia, **3** Hawassa University, Hawassa, Ethiopia, **4** St. Peter Specialized Hospital, Addis Ababa, Ethiopia, **5** Nossal Institute, Melbourne School of Population and Global Health, University of Melbourne, Victoria, Australia

* amteklu@merqconsultancy.com

**Data Availability Statement:** The data management team at MERQ Consultancy will make the data available upon request. Setegn Tigabu

## Abstract

### Introduction

Providing patient-centered care is one of the key focus areas of the Ethiopian Health Service Transformation Plan. To this end, improving health literacy of the community is critical. However, there is limited evidence about the health literacy of Ethiopians, especially among those who visit health facilities.

### Objective

The aim of this study was to examine awareness of diagnosis, treatment plan and prognosis among patients at the time of their exit from public hospitals and health centers.

### Methods

A cross-sectional study was conducted among 627 patients in two public hospitals and selected health centers in Addis Ababa, using a systematic random sampling technique from inpatient and outpatient departments (OPD). A total of 579 study participants had complete data and were included in this analysis. A structured, pre-tested and interview-administered questionnaire was used to collect data. We used proportions to describe the findings and logistic regression analyses to assess factors associated with awareness of diagnosis, treatment plan and prognosis.

### Result

About three-fifths (61.9%) and 52.8% of the study participants knew correctly their diagnosis and treatment plan respectively. More than two-thirds, 68.4%, said that they knew about the prognosis of their illness. However, only 21 (3.6%) patient medical records had information on prognosis. Gynecologic patients had significantly lower awareness about their diagnosis

(setegn.t@merqconsultancy.org), the data manager/data protection manager at MERQ consultancy will be responsible for sharing the dataset.

**Funding:** This study was funded by MERQ PLC.

**Competing interests:** No competing interests were disclosed.

and treatment plan as compared to those from a general outpatient department. Emergency patients had significantly lower awareness of their treatment plan (OR = 0.27; 95% CI: 0.11,0.68) and prognosis (OR = 0.21; 95% CI: 0.09,0.50) than new OPD patients. Patients who indicated they had a good experience at their clinical assessment had significantly lower awareness of their prognosis (OR = 0.25; 95% CI: 0.08, 0.81).

## Conclusion

A significant proportion of patients didn't know their correct diagnosis, treatment plan and prognosis. This was more pronounced among gynecologic and emergency patients. More efforts are needed to strengthen patient-provider interaction.

## Introduction

Patient care is shifting globally from a traditional approach towards a patient-centered approach that involves patients in their own care [1,2]. Patient-centered care is believed to optimize patients' health literacy by ensuring their access to information about their diagnosis, treatment and prognosis. Improved health literacy will help patients understand, appraise and use health information to make decisions relevant to their health condition [3]. A patient-centered approach is directly related to effectiveness of patient care, increased patient satisfaction [4,5], better patient involvement in disease prevention, improved patient skills in self-management [6,7], patient engagement, and perceived quality of care [8,9].

Effectiveness of self-management of disease depends on close communication between healthcare providers and patients [10]. Patients' health literacy and self-efficacy to manage their disease is also highly dependent on patient-provider communication and patient involvement in decision-making about their treatment plan [11,12], both of which are integral components of patient-centered care. Consequently, initiatives in patient-centered care and health literacy are contributing to the improvement of patients' skills in self-management of their own diseases [2,5].

A systematic review on patient involvement reported that patients can actively monitor their own care if they get adequate information and are actively involved in the decision-making process [12]. Research has also showed that patient involvement in their own care and better patient-provider communication are strongly associated with medication adherence [13,14]. However, patient-provider communication is poor in many settings. As a result, only a limited proportion of patients receive adequate information about their health and healthcare. For instance, a meta-analysis studies, mostly from high-income countries, conducted among cancer patients reported that only 49% were aware of their prognosis [15].

Patients' involvement in their medical care will promote mutual accountability and understanding between patients and health care providers. Primary care providers are ideally placed to engage patients in a discussion about their health conditions, treatment plan and lifestyle changes. Well-informed patients are more likely to feel confident to report both positive and negative experiences about their health and illness [16]. However, even if patients have a right to adequate information about their clinical assessment procedure, diagnosis and treatment plan and prognosis, evidence suggests that most patients do not know about their right [17].

Awareness about diagnosis, treatment plan and prognosis are key elements of patient-centered care. Evidence from another systematic review showed that about 75% of patients were not aware of their prognosis and 96% were not aware of their diagnosis [18]. Another study

has shown that the majority (97.4%) of patients knew their physician's name. The same study reported that men have better awareness about their health condition, treatment complication, medication administrated and plan of care than women [19]. Similarly, a study of awareness of prognosis in oncological patients at the end of life showed that the large majority of terminal cancer patients did not have adequate information about their diagnosis and prognosis [20]. On the other hand, research indicated that many patients with early-stage cancer want detailed prognostic information, presented in an open and honest manner [21].

With the ultimate aim of providing effective care for all patients in Ethiopia, the patient-centered approach has been integrated in the Health Sector Transformation Plan (HSTP) [22]. Published evidence from the Tigray region of Ethiopia showed that a considerable proportion of patients had poor experiences in their medical care [23]. A qualitative study in the Southern region of Ethiopia indicated that patients have positive perception of patient-centered care [24].

However, there is limited evidence on patients' awareness about their health condition and management of their disease. Besides, the existing evidence is largely based on patients' verbal reports. To our knowledge, there is no published study that objectively measures patients' awareness of their diagnosis, prognosis and treatment plan. Therefore, the aim of the current study was to examine awareness of diagnosis, treatment plan and prognosis among patients attending public hospitals and health centers in Addis Ababa, Ethiopia.

## Methods

### Study design and context

An institution-based cross-sectional study was conducted to assess awareness of diagnosis, treatment plan, and prognosis among adult patients. St. Paul Hospitals Millennium Medical College (SPHMMC) and St. Peter Specialized Hospital and their catchment health centers were the study areas. The study was conducted between July 20th and August 30th of 2019. Patients who visited the outpatient department and inpatient departments of internal medicine, surgery, and gynecology and specialty clinics were listed. Patients who were critically ill and unable to respond and who visited the antenatal care and delivery unit and discharged from obstetrics admission, family planning clients, and patients at the HIV treatment clinic were not included in this study.

### Sampling of study participants

Using proportion of effect (p) 55%, 95% level of significance, 5% margin of error 1.5 design effect and 10% non-response rate, we needed 627 study participants. The two public hospitals were selected purposively. A simple random sampling technique was used to select the catchment health centers of the two public hospitals and a systematic random sampling technique was used to recruit the study participants from the selected health institutions. The total sample was proportionally allocated to the health institutions based on their patient load. The sampling fraction was determined based on the daily patient flow of the health institution.

### Inclusion/Exclusion criteria

All adult patients who would visit the outpatient and inpatient departments of Internal Medicine, Surgery, Gynaecology, Emergency and Specialty Clinics of the study facilities were eligible for inclusion. Patients who were critically ill and unable to respond were not eligible for this study. Besides, patients who visited ANC, Delivery, Family Planning, Anti-Retroviral treatment clinics were excluded.

## Data collection procedures and instruments

After training of the data collectors, the data-collection process and questionnaire were pre-tested in similar contexts. Refinements to the data-collection process and questionnaire were made based on the lessons from the pre-test. Members of the research team interviewed study participants at the time of their exit or discharge from each facility. The data-collection process was supervised by two researchers. Prior to conducting interviews, the data collectors obtained informed consent from each study participant.

We used a structured and interview-administered questionnaire to collect data. The questionnaire had six domains: demographic characteristics, Patient Experience Questionnaire (PEQ) (25), Patient Perception of Patient-Centeredness (PPPC) scale (26), Perception on Quality of Health Service, General Patient Satisfaction Scale (27) and Items on Patients Awareness on their Diagnosis, Treatment and Prognosis (28).

The PEQ scale was used to explore patients' experience of the care they received. Except for waiting time and perceived benefits, all items were based on 5-point response scale: "1 = not at all", "2 = small extent", "3 = moderate extent", "4 = large extent", and "5 = very large extent". The PPPC was used to measure patient perceptions of patient-centered care during their visit to the health facilities. The instrument had 14 items scored on a 4-point Likert scale ranging from completely agree on the idea to not at all, and no subscales, with Cronbach's reliability for the global score of 0.71(36). Then, experiences of patients, patient perception of patient-centeredness of care and general patient satisfaction were classified in to good and poor or satisfied and not satisfied using 75% of as a cut-off point [25–27].

Information related to the outcome variables, such as patients' diagnosis, treatment plan, and prognosis, were collected from both the patient and the patient's medical record by two different data collectors. Knowledge about prognosis of the disease was determined using awareness about the expected outcome of the treatment, which included cure, chronicity, and threat to life. Agreement between the two sources of information was examined [28]. Differences between the two data sources were verified through a thorough discussion with three experienced physicians. Patients were classified as having awareness when information from patients' verbal report agreed with the information from their medical record.

## Data analysis

Data were entered, cleaned and coded using Epi-data software. We used Stata 15.0 for statistical analysis. We summarized descriptive information in tables using proportions. We used logistic regression models to assess factors associated with awareness of diagnosis, treatment plan and prognosis. In the analysis of factors associated with awareness of diagnosis, treatment and prognosis, we controlled for the effects of potential confounders including type of facilities and departments within facilities. We used multiple logistic regression models to identify factors associated with each of awareness of diagnosis, treatment and prognosis. As the sample is proportionally distributed between hospitals and health centers, we don't expect significant effect of clustering. We presented measures of adjusted odds ratios with 95% confidence interval and p values. P values less than 0.05 were considered to be statistically significant.

## Ethical consideration

Ethical clearance to conduct this study was obtained from the Institutional Review Board of SPHMMC, St. Peter's Specialized Hospital Ethical Review Committee Office (ERCO). Written informed consent was obtained from all study participants.

## Results

### Socio-demographic and clinical characteristics of the respondents

A total of 627 study participants were interviewed, making the response rate 98.1%. From these, 579 (92.3) had complete data and were included in this analysis. The mean age of the respondents was 40.3 (SD = 16.3) years. The majority were housewives (27.8%), orthodox Christians (78.9%), and had Amharic as their mother tongue (63.9%) (see Table 1).

Among the study participants, 48.5% received the service at a general outpatient department. New outpatient visitors represented 56.3% and 67.4% had their first encounter with the clinician. A little more than half, 55.8%, felt very unwell and 52.2% were very much worried

**Table 1. Socio-demographic and clinical characteristics of study participants (n = 579).**

| Variables | Categories | Values |
|---|---|---|
| Age (mean, SD) | | 40.3 (16.3) |
| Female (n, %) | | 339 (58.5%) |
| Urban residence (n, %) | | 494 (85.3) |
| Currently married (n, %) | | 350 (60.4) |
| Employment (n, %) | Government or NGO Employed | 141 (24.4) |
| | Other occupation* | 277 (47.8) |
| Education (n, %) | None | 112 (19.3) |
| | Able to read and write | 81 (14.0) |
| | Primary (1–8) | 175 (30.2) |
| | Secondary & Preparatory (9–12) | 163 (28.2) |
| | Tertiary or Higher | 48 (8.3) |
| Religion (n, %) | Orthodox Christian | 457 (78.9) |
| | Muslim | 80 (13.8) |
| | Other religious followers** | 42 (7.3) |
| Mother tongue is Amharic (n, %) | | 370 (63.9) |
| First encounter with the clinician | | 390(67.4) |
| Visited hospital | | 325(56.1) |
| Department | General OPD | 281(48.5) |
| | Internal Medicine | 75(13) |
| | Surgery | 80(13.8) |
| | Gynecology | 83(14.3) |
| | Other departments*** | 60(10.4) |
| Care type | Outpatient (new) | 326(56.3) |
| | Inpatient | 95(16.4) |
| | Emergency | 41(7.1) |
| | Outpatient (Follow-up) | 117(20.2) |
| Feeling | Very unwell | 323(55.8) |
| | Moderately unwell | 221(38.2) |
| | Slightly unwell / well | 35(6) |
| Worry about illness | Very worried | 302(52.2) |
| | Moderately worried | 222(38.3) |
| | Slightly/not worried | 55(9.5) |

*Farmer, merchant, self-employed, retired

**Protestant, Catholic, Wakefeta

***Emergency& specialty clinics.

about their problem when they came to the health facility. Of all the study participants, 56.1% received service at hospital and 72.9% of them arrived at the health institution before 6:30 PM of local time (see Table 1).

### Experience and perception of patients

Of the total study participants, 53.4% were not satisfied by the service they received and 41.3% had a poor experience during the contact for their clinical assessment. Also 39.2% had a poor perception of patient-centeredness and 21.8% had a poor perception of quality of service. A significant majority, 89.1%, mentioned that the physician didn't introduce himself/herself to them during the clinical assessment and 92.6% did not know the name of the physician who treated them. Close to half, 44.6%, didn't know the career position of the health provider. About a third, 34.7% indicated that the clinician did not talk to them in their mother tongue.

### Awareness of diagnosis, treatment plan and prognosis

Among the study participants, 21.6% indicated that their clinician didn't inform them about their diagnosis and 80.3% responded that they knew their diagnosis. However, 38.1% of them didn't know their correct diagnosis at the time of their exit from the health facility.

More than four out of five patients, 82.2%, reported that they have received adequate information while 85.1% responded that they were informed about their treatment plan. About 92% of the study participants said they know about their treatment plan. Even if three quarters had no chance to discuss with the care provider after they bought the medication, 72.7% obtained the prescribed medication within the health facility. During verification of patients' verbal report with the treatment plan written on the patient medical record, we found that 47.2% did not know their treatment plan correctly.

From the study participants, 58.5% reported that they were told about the prognosis of their illness by the clinician. More than two-thirds, 68.4%, said that they knew about the prognosis of their illness. However, only a fifth of patient medical records had information on prognosis status, of which 13 (62%) had similar information as described by the patients.

Among the study participants, more than half mentioned that their illness needed a follow-up. Of these, close to half were told about their follow-up date by their clinician and 46.8% knew their follow-up date. However, most of the patients' medical records had no written information about the follow-up date. Of the 75 patient cards with written follow up date, nearly two-thirds had the same information as that of the patients' verbal report. Details of these are illustrated in Table 2.

### Factors associated with awareness of diagnosis, treatment and prognosis

Patients who visited the Gynaecology department had significantly lower awareness about their diagnosis and treatment plan, while surgical patients had significantly lower awareness of their treatment plan and prognosis. Patients treated at the emergency department had significantly lower awareness of their treatment plan and prognosis. On the other hand, patients with good perception of patient-centeredness of care had significantly higher awareness of their prognosis. Details are shown in Table 3 below.

## Discussion

In this facility-based cross-sectional study, we assessed patients' awareness about their diagnosis, treatment plan and prognosis using data from exit interviews and patient medical records. The finding showed that the level of patients' awareness about their diagnosis was considerably

**Table 2. Awareness of diagnosis, treatment plan and prognosis.**

| Background variables | Categories | Aware of diagnosis n (%) | P value | Aware of treatment plan n (%) | P value | Aware of prognosis n (%) | P value |
|---|---|---|---|---|---|---|---|
| Gender | Male | 120(60) | 0.469 | 110(50.9) | 0.469 | 152(63.3) | 0.028 |
| | Female | 179(63.3) | | 163(54.2) | | 244(72) | |
| Age | 35 and below Years | 144(61) | 0.695 | 134(53.2) | 0.869 | 209(74.9) | 0.001 |
| | Above 35 Years | 155(62.8) | | 139(52.5) | | 187(62.3) | |
| Marital status | Currently married | 190(63.8) | 0.287 | 175(57.4) | 0.012 | 236(67.4) | 0.537 |
| | Currently not married | 109(58.9) | | 98(46.2) | | 160(69.9) | |
| Educational status | None | 49(59.8) | 0.983 | 40(42.1) | 0.111 | 62(55.4) | 0.004 |
| | Able to read & write | 47(63.5) | | 41(59.4) | | 55(67.9) | |
| | Primary (1–8) | 96(62.3) | | 86(53.1) | | 130(74.3) | |
| | Secondary (9–12) | 80(61.1) | | 77(53.1) | | 110(67.5) | |
| | Tertiary or Higher | 27(64.3) | | 29(63) | | 39(81.3) | |
| Service providing department | General OPD | 146(62.4) | 0.061 | 156(58.4) | 0.036 | 235(83.6) | <0.001 |
| | Internal Medicine | 41(66.1) | | 36(56.3) | | 29(38.7) | |
| | Surgery | 41(59.4) | | 31(44.9) | | 46(57.5) | |
| | Gynecology | 31(48.4) | | 29(43.9) | | 64(77.1) | |
| | Emergency/specialty | 40(74.1) | | 21(41.2) | | 22(36.7) | |
| Patient getting care as | Outpatient (new) | 157(61.1) | 0.501 | 169(56.7) | <0.001 | 254(77.9) | <0.001 |
| | Inpatient | 48(57.1) | | 35(41.2) | | 54(56.8) | |
| | Emergency | 21(61.8) | | 8(23.5) | | 14(34.2) | |
| | Outpatient (repeat) | 73(67.6) | | 61(61) | | 74(63.3) | |
| Encounter with the clinician | First Contact | 180(57.9) | 0.014 | 170(48.6) | 0.005 | 274(70.3) | 0.166 |
| | More than one time | 119(69.2) | | 103(61.7) | | 122(64.6) | |
| Residence | Urban | 266(63.2) | 0.132 | 239(53.8) | 0.250 | 350(70.9) | 0.002 |
| | Rural | 33(53.2) | | 34(46.6) | | 46(54.1) | |
| Experience at examination | Poor | 131(60.4) | 0.530 | 120(47.6) | 0.021 | 169(60.4) | <0.001 |
| | Good | 168(63.2) | | 153(57.7) | | 227(75.9) | |
| Patient-centeredness | Poor | 107(59.4) | 0.391 | 101(47.9) | 0.062 | 132(55.2) | <0.001 |
| | Good | 192(63.4) | | 172(56.2) | | 264(77.7) | |
| Quality of Service | Poor | 53(60.9) | 0.834 | 53(49.1) | 0.383 | 69(54.8) | <0.001 |
| | Good | 246(62.1) | | 220(53.8) | | 327(72.2) | |
| Patient Satisfaction | Unsatisfied | 168(64.9) | 0.150 | 157(56.5) | 0.071 | 230(74.4) | 0.001 |
| | Satisfied | 131(58.5) | | 116(48.5) | | 166(61.5) | |
| Used mother tongue | No | 110(66.3) | 0.153 | 92(52) | 0.786 | 125(62.2) | 0.019 |
| | Yes | 189(59.6) | | 181(53.2) | | 271(71.7) | |
| Health Facility type | Hospital | 173(62.9) | 0.601 | 139(51.5) | 0.529 | 191(58.8) | <0.001 |
| | Health Center | 126(60.6) | | 134(54.3) | | 205(80.7) | |

low. This was consistent with previous studies in Shanghai, China [28], United Kingdom [29] and Sri Lanka [30]. The lower level of awareness might relate to patients' awareness of their right to ask for information about their health condition. In this regard, patients' awareness about their rights to ask for and receive information about their own health condition and treatment plans need to be improved. Besides, clinicians also have a responsibility to ensure that patients are well informed about their medical care [31].

Our study found a relatively higher level of awareness of diagnosis than reported by some other countries [31–35]. These other studies recruited patients who had similar illnesses, but we sampled patients from various departments and health institutions. On the other hand,

**Table 3. Factors associated awareness of diagnosis, treatment plan and prognosis from multiple logistic regression models.**

| | Awareness of Diagnosis | | Awareness of Treatment | | Awareness of Prognosis | |
|---|---|---|---|---|---|---|
| | OR* (95% CI) | P | OR* (95% CI) | P | OR* (95% CI) | P |
| Age (Ref: <36 years) | | | | | | |
| Age>35 years | 0.80(0.51,1.27) | 0.349 | 0.85(0.55,1.31) | 0.466 | 0.81(0.5,1.32) | 0.406 |
| Gender (Ref: Male) | | | | | | |
| Female | 1.40(0.91,2.15) | 0.129 | 1.20(0.8,1.81) | 0.374 | 1.17(0.73,1.86) | 0.511 |
| Marital status (Ref: Currently married) | | | | | | |
| Currently not married | 0.80(0.52,1.22) | 0.293 | 0.66(0.44,1) | 0.049 | 1.01(0.64,1.59) | 0.983 |
| Educational status (Ref: None) | | | | | | |
| Able to read and write | 1.03(0.50,2.10) | 0.938 | 1.51(0.74,3.07) | 0.257 | 1.57(0.73,3.37) | 0.246 |
| Primary (1–8) | 1.13(0.61,2.12) | 0.694 | 1.54(0.84,2.8) | 0.160 | 1.77(0.93,3.36) | 0.083 |
| Secondary & Preparatory (9–12) | 1.06(0.56,2.01) | 0.853 | 1.69(0.91,3.12) | 0.096 | 1.29(0.67,2.46) | 0.445 |
| Tertiary or Higher | 1.24(0.54,2.86) | 0.617 | 2.56(1.14,5.72) | 0.022 | 2.71(1.02,7.21) | 0.045 |
| Department (Ref: General OPD) | | | | | | |
| Internal Medicine | 0.74(0.32,1.68) | 0.468 | 0.54(0.23,1.27) | 0.159 | 0.11(0.05,0.27) | <0.001 |
| Surgery | 0.57(0.25,1.35) | 0.202 | 0.34(0.14,0.84) | 0.019 | 0.34(0.14,0.84) | 0.020 |
| Gynecology | 0.29(0.12,0.73) | 0.009 | 0.30(0.12,0.77) | 0.013 | 0.92(0.34,2.52) | 0.879 |
| Other | 1.32(0.54,3.25) | 0.540 | 0.37(0.15,0.93) | 0.034 | 0.20(0.08,0.49) | <0.001 |
| Type of care (Ref: New outpatient) | | | | | | |
| In patient | 1.33(0.68,2.59) | 0.407 | 0.71(0.37,1.35) | 0.293 | 0.61(0.31,1.18) | 0.139 |
| Emergency (Emergency Room) | 0.82(0.34,1.98) | 0.661 | 0.27(0.11,0.68) | 0.006 | 0.21(0.09,0.50) | <0.001 |
| Outpatient (Follow-up) | 0.99(0.50,1.96) | 0.975 | 1.09(0.55,2.16) | 0.806 | 0.72(0.35,1.47) | 0.361 |
| Encounter (Ref: First) | | | | | | |
| Repeated | 1.69(0.96,2.99) | 0.071 | 1.82(1.05,3.16) | 0.034 | 1.39(0.75,2.55) | 0.293 |
| Residence (Ref: Urban) | | | | | | |
| Rural | 0.65(0.35,1.22) | 0.183 | 0.91(0.5,1.68) | 0.774 | 0.68(0.37,1.27) | 0.226 |
| Patient Experience (Ref: Poor) | | | | | | |
| Good | 0.84(0.39,1.8) | 0.658 | 1.63(0.8,3.34) | 0.178 | 0.25(0.08,0.81) | 0.020 |
| Patient-centeredness of service (Ref: poor) | | | | | | |
| Good | 1.38(0.63,3.06) | 0.423 | 0.92(0.44,1.92) | 0.829 | 10.21(3.11,33.55) | <0.001 |
| Perceived quality of service (Ref: Poor) | | | | | | |
| Good | 0.89(0.51,1.58) | 0.701 | 0.85(0.51,1.43) | 0.542 | 1.19(0.69,2.06) | 0.525 |
| General satisfaction (Ref: Not satisfied) | | | | | | |
| Satisfied | 0.68(0.43,1.05) | 0.082 | 0.75(0.49,1.14) | 0.179 | 0.75(0.47,1.19) | 0.222 |
| Used mother tongue (Ref: No) | | | | | | |
| Yes | 0.75(0.49,1.16) | 0.194 | 1(0.66,1.51) | 0.998 | 0.98(0.62,1.54) | 0.934 |
| Facility type (Ref: Hospital) | | | | | | |
| Health Center | 0.59(0.3,1.17) | 0.129 | 0.45(0.22,0.94) | 0.033 | 0.78(0.36,1.69) | 0.532 |

*ORs are adjusted for all the variables in the first column of this table.

compared with studies conducted in South Africa [36] and United States [37], our study found a relatively low level of awareness of diagnosis. This might be due to the recruitment of patients with chronic illness for those studies. Patients with chronic illness usually have better chances to frequently contact their physicians and have more time to ask and acquire detailed information. This would help them to improve their awareness about their diagnosis, treatment plan and prognosis [36].

Even though it was higher than has been reported in other settings, we found a low level of patient awareness about their treatment plan and prognosis [17,20]. On the other hand, relatively higher levels of awareness about treatment plans were reported in South Africa [36] and China [38]. The difference might be related to the variations in medical treatment approach and better provider-patient interaction. The differences in the health systems' capacity to deliver patient-centered services could also explain for these differences.

Low level of awareness of diagnosis among patients who received service at the gynecology department and among those who contacted the clinician for the first time might partly be due to their limited exposure to the healthcare system. This limited exposure may reduce the chances to seek and access more information about their illness. In China, adequate knowledge about chronic diseases was strongly associated with regular check-ups, especially for those who attended hospital settings [35].

Patients who received a service as an emergency patient had lower awareness of their treatment plan and prognosis compared to other outpatient cases. This is consistent with the study findings from Sudan that indicated emergency patents are necessarily not as well informed as others [39]. Consistent with the current study findings, patients who received care as an emergency patient and received service at general outpatient departments and gynecology departments had lower awareness of the prognosis of their illness. Visiting specialty clinics (like diabetes and hypertension clinics) and having repeated check-ups seem to be associated with a higher level of knowledge about current condition of the illness. Further, those who had a poor experience during the examination had a low level of awareness about the prognosis of their illness.

A significant proportion of patients didn't know their correct diagnosis, treatment plan and prognosis. This was more pronounced among gynecologic and emergency patients. This shows that the health system in Addis Ababa needs to devise strategies to improve the quality of care provided in hospitals and health centers, particularly in terms of ensuring patients get all the necessary information about their diagnosis, treatment and prognosis. In this regard, improving patient-provider interaction needs to be one of the focus areas, especially for patients visiting gynecology, emergency and surgery departments. Further studies, with higher sample size and geographic coverage, are needed to explore the actual factors that contributed to the gaps in awareness of diagnosis, treatment plan and prognosis among patients in Addis Ababa.

There are some limitations associated with this study. The first limitation of this study is its cross-sectional nature. It was not possible to establish the direction of effect of some outcomes and explanatory variables. Secondly, this study focused on the main diagnosis. It didn't fully explore information related to other comorbid conditions. Thirdly, information on how clinicians provide information to their patients was also not part of this study. Besides, the findings of this study may not reflect awareness of patients who visit private hospitals and clinics. Finally, there was limited information about prognosis in patient medical records and it was not possible to objectively verify patients' verbal reports about their prognosis.

## Conclusion

This study has shown that about two out of five patients didn't know their diagnosis correctly. About half of the patients didn't know their treatment plan correctly and one-third of them didn't know their prognosis. Gynecologic and surgical patients had significantly lower awareness about their diagnosis and treatment plan. Emergency patients had significantly lower awareness of their treatment plan and prognosis. Patients who reported a good experience at their clinical assessment had significantly lower awareness of their prognosis. Health facilities

need to improve patient-provider interaction and ensure that patients receive all important information, presented in a way they can understand.

## Acknowledgments

The authors would like to acknowledge the MERQ for funding this research. The authors also appreciate the respective hospitals and health centers and the study participants for their help in providing the necessary information for successful completion of this study.

## Author Contributions

**Conceptualization:** Alula M. Teklu, Mebratu Abraha, Tegenne Legesse, Frehiwot Belachew.

**Data curation:** Alula M. Teklu, Mebratu Abraha, Tegenne Legesse, Bizuayehu Aseffa, Frehiwot Belachew.

**Formal analysis:** Alula M. Teklu, Abrham Getachew, Bizuayehu Aseffa, Million Molla, Tilahun N. Haregu.

**Funding acquisition:** Alula M. Teklu.

**Investigation:** Alula M. Teklu, Frehiwot Belachew.

**Methodology:** Alula M. Teklu, Mebratu Abraha, Tegenne Legesse, Mahteme Bekele, Abrham Getachew, Million Molla, Frehiwot Belachew, Tilahun N. Haregu.

**Resources:** Frehiwot Belachew.

**Software:** Alula M. Teklu, Tegenne Legesse, Tilahun N. Haregu.

**Supervision:** Alula M. Teklu, Mebratu Abraha, Mahteme Bekele, Abrham Getachew, Bizuayehu Aseffa, Million Molla, Frehiwot Belachew.

**Validation:** Mebratu Abraha, Million Molla.

**Visualization:** Alula M. Teklu, Mahteme Bekele.

**Writing – original draft:** Alula M. Teklu, Mebratu Abraha, Tegenne Legesse, Mahteme Bekele, Abrham Getachew, Bizuayehu Aseffa, Tilahun N. Haregu.

**Writing – review & editing:** Alula M. Teklu, Mebratu Abraha, Tegenne Legesse, Mahteme Bekele, Bizuayehu Aseffa, Million Molla, Frehiwot Belachew, Tilahun N. Haregu.

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
