## [Decision Letter · Decision Letter 0]

6 May 2021

PONE-D-20-36374

Awareness of diagnosis, treatment plan and prognosis among patients attending public hospitals and health centers in Addis Ababa, Ethiopia.

PLOS ONE

Dear Dr. Teklu,

Thank you for submitting your manuscript to PLOS ONE. After careful consideration, we feel that it has merit but does not fully meet PLOS ONE’s publication criteria as it currently stands. Therefore, we invite you to submit a revised version of the manuscript that addresses the points raised during the review process.

Your manuscript has undergone the peer-review process and the reviewers have provided their comments/suggestions. Kindly address these points/concerns before we make a decision.

We look forward to receiving your revised manuscript.

Kind regards,

Kingston Rajiah

Academic Editor

PLOS ONE

Journal Requirements:

3. Please include additional information regarding the survey or questionnaire used in the study and ensure that you have provided sufficient details that others could replicate the analyses. For instance, if you developed a questionnaire as part of this study and it is not under a copyright more restrictive than CC-BY, please include a copy, in both the original language and English, as Supporting Information, or include a citation if it has been published previously.

4. In the Methods, please discuss whether and how the questionnaire was validated for your context and/or pre-tested. If these did not occur, please provide the rationale for not doing so.

5. In your statistical analyses, please state whether you accounted for clustering by locality. For example, did you consider using multilevel models?

6. In statistical methods, please refer to any post-hoc corrections to correct for multiple comparisons during your statistical analyses. If these were not performed please justify the reasons. Please refer to our statistical reporting guidelines for assistance (https://journals.plos.org/plosone/s/submission-guidelines.#loc-statistical-reporting).

7. Thank you for stating the following in the Financial Disclosure section:

[This study was funded by MERQ PLC.]. 

We note that one or more of the authors have an affiliation to the commercial funders of this research study: MERQ consultancy PLC

8. We note that you have indicated that data from this study are available upon request. PLOS only allows data to be available upon request if there are legal or ethical restrictions on sharing data publicly. For information on unacceptable data access restrictions, please see http://journals.plos.org/plosone/s/data-availability#loc-unacceptable-data-access-restrictions.

9. We note you have included a table to which you do not refer in the text of your manuscript. Please ensure that you refer to Table 2 in your text; if accepted, production will need this reference to link the reader to the Table.

10. Please include your tables as part of your main manuscript and remove the individual files. Please note that supplementary tables (should remain/ be uploaded) as separate "supporting information" files.

Reviewers' comments:

Reviewer's Responses to Questions

**Comments to the Author**

1. Is the manuscript technically sound, and do the data support the conclusions?

Reviewer #1: Partly

Reviewer #2: Partly

2. Has the statistical analysis been performed appropriately and rigorously? 

Reviewer #1: Yes

Reviewer #2: No

3. Have the authors made all data underlying the findings in their manuscript fully available?

Reviewer #1: Yes

Reviewer #2: No

4. Is the manuscript presented in an intelligible fashion and written in standard English?

Reviewer #1: Yes

Reviewer #2: Yes

5. Review Comments to the Author

Reviewer #1: 1. An overall comment is for the quality of writing: English language is sometimes not the most appropriate. I urge authors to check the text thoroughly before resubmission and to read other published papers from the journal to check the style of writing.

2. The author must add the inclusion and exclusion criteria in the study.

3. The research work Awareness of diagnosis, treatment plan and prognosis among patients attending public hospitals and health centers in Addis Ababa, Ethiopia is well conducted among the patients, the sample size is to be increased, and further work may be carried out.

4. The authors also add the limitations of the study.

Reviewer #2: General comment: this study examines an important question about patients perception and understanding of care received. Inconsistencies in data and need of further clarity on methodology need to be examined.

Specifics:

Abstract: “Patients with good experience at their clinical assessment had significantly lower awareness of their prognosis"

Conclusion: “Patients with good experience at their clinical assessment had significantly higher awareness of their prognosis.”

 both statements are in contradiction, also "(OR=0.25; 95% CI: 0.08, 0.81)” in the abstract does not correspond to the data in the table

Did the authors collect data on gender of health care providers? Was there a difference in patient knowledge of diagnosis, treatment plan or prognosis depending on provider gender? If provider gender was not examined would perhaps rephrase: “Patient care is shifting globally towards patient-centered approach [1, 2].”

“Questionnaire (PEQ) (25), Patient Perception of Patient-Centeredness (PPPC) scale (26), Perception on Quality of Health Service, General Patient Satisfaction Scale (27) and Items on Patients Awareness on their Diagnosis, treatment and Prognosis (28).”

was this translated or did the interviewer use the English version and each translate themselves? Were these instruments validated in the translated language? What language was used? Please provide a copy of the English and translated version as an appendix. Some validation of the instruments would be useful for this and future publications. This will also help readers to better understand how variables such as "patient centeredness" or "prognosis" or the variarious satisfaction/good care variables were defined.

Also authors could provide a reference indicating that documenting prognosis separate from the diagnosis and treatment plan is a requirement in medical notes. Is this now a requirement in Ethiopia, WHO, etc for example? Often prognosis is implicit in the diagnosis and treatment plan in medical notes.

“We used logistic regression models to assess factors associated with awareness of diagnosis, treatment plan and prognosis”

What variables were included in these models? Has were interactions assessed and managed? It is not clear in the results section which results were bivariate, which were from multivariate models and or variables were included in the models.

Gynecologic patients had significantly lower awareness about their diagnosis and treatment plan, compared to who? Why is this different from females in general?

How could patients understand prognosis if they don’t know the diagnosis or treatment plan?

Also in the introduction would indicate the location of the references like in the discussion.

% in the text of the result would be better in a table.

Data from the tables to keep tract

Awareness of Diagnosis Awareness of Treatment Awareness of Prognosis

Female

179(63.3) 163(54.2) 244(72)

1.40(0.91,2.15) 0.129 1.20(0.8,1.81) 0.374 1.17(0.73,1.86)0.511

Gynecology

31(48.4) 29(43.9) 64(77.1)

0.29(0.12,0.73) 0.009 0.30(0.12,0.77) 0.013 0.92(0.34,2.52) 0.879

General satisfaction (Ref: Not satisfied)

131(58.5) 116(48.5) 166(61.5)

0.68(0.43,1.05) 0.082 0.75(0.49,1.14) 0.179 0.75(0.47,1.19) 0.222

6. PLOS authors have the option to publish the peer review history of their article (what does this mean?). If published, this will include your full peer review and any attached files.

Reviewer #1: No

Reviewer #2: No

---

## [Author Response · Author response to Decision Letter 0]

24 Oct 2021

Date Jun 5, 2021

Kingston Rajiah

Academic Editor

PLOS ONE

Re: Awareness of diagnosis, treatment plan and prognosis among patients attending public hospitals and health centers in Addis Ababa, Ethiopia.

Dear Dr. Kingston Rajiah,

Thank you for review of our manuscript. We found your comments and reviewers’ comments very helpful. We have revised the manuscript based on your comments. We have attached:

1. Response to Reviewers - A rebuttal letter that responds to each point raised by the academic editor and reviewer(s). 

2. Revised Manuscript with Track Changes - A marked-up copy of your manuscript that highlights changes made to the original version. 

3. Manuscript - An unmarked version of your revised paper without tracked changes. 

We look forward to receiving hearing from you.

Kind regards,

Dr Alula M Teklu 

On behalf of the authors 

Responses to Editor’s comments

Response: We have edited the title page and file names based on the PLOS ONE’s style requirements.

Response: The manuscript has now been copyedited for language usage by Christopher Crompton who is an experienced editor. 

Response: We have included these files as per the PLOS ONE guidelines

3. Please include additional information regarding the survey or questionnaire used in the study and ensure that you have provided sufficient details that others could replicate the analyses. For instance, if you developed a questionnaire as part of this study and it is not under a copyright more restrictive than CC-BY, please include a copy, in both the original language and English, as Supporting Information, or include a citation if it has been published previously.

Response: English and Amharic version of the questionnaire and the consent forms that we developed as part of this study are attached as supplemental file.

4. In the Methods, please discuss whether and how the questionnaire was validated for your context and/or pre-tested. If these did not occur, please provide the rationale for not doing so.

Response: After training of the data collectors, the data collection process and questionnaire were pre-tested in similar contexts. Refinements to the data collection process and questionnaire were made based on the lessons from the pre-test. This information is included in the revised version

5. In your statistical analyses, please state whether you accounted for clustering by locality. For example, did you consider using multilevel models?

Response: In the analysis of factors associated with awareness of diagnosis, treatment and prognosis, we controlled for the effects of potential confounders including type of facilities and departments within facilities. As the sample is proportionally distributed between hospitals and health centers, we don’t expect significant effect of clustering.

6. In statistical methods, please refer to any post-hoc corrections to correct for multiple comparisons during your statistical analyses. If these were not performed please justify the reasons. Please refer to our statistical reporting guidelines for assistance (https://journals.plos.org/plosone/s/submission-guidelines.#loc-statistical-reporting).

 Response: Most of our variables are categorical. Post-hoc multiple comparisons were not needed for these types of analysis. In the logistic regression models, we compared each category against the reference category.

7. Thank you for stating the following in the Financial Disclosure section:

[This study was funded by MERQ PLC.]. 

We note that one or more of the authors have an affiliation to the commercial funders of this research study: MERQ consultancy PLC

Response: The authors’ contribution section is amended as suggested.

Response: The funding statement is amended as suggested.

Response: The competing interests statement is amended.

Response: Updated Funding Statement and Competing Interests Statement are included in the updated cover letter

8. We note that you have indicated that data from this study are available upon request. PLOS only allows data to be available upon request if there are legal or ethical restrictions on sharing data publicly. For information on unacceptable data access restrictions, please see http://journals.plos.org/plosone/s/data-availability#loc-unacceptable-data-access-restrictions.

Response: Data contain potentially identifying patient information. Requests for access to data can be submitted to the corresponding author who can process data access.

9. We note you have included a table to which you do not refer in the text of your manuscript. Please ensure that you refer to Table 2 in your text; if accepted, production will need this reference to link the reader to the Table.

Response: Included. 

10. Please include your tables as part of your main manuscript and remove the individual files. Please note that supplementary tables (should remain/ be uploaded) as separate "supporting information" files.

Response: All tables are within the main manuscript

Reviewers' comments:

Comments to the Author

Reviewer #1: 

1. An overall comment is for the quality of writing: English language is sometimes not the most appropriate. I urge authors to check the text thoroughly before resubmission and to read other published papers from the journal to check the style of writing.

Response: The language has been carefully edited by Christopher Crompton.

2. The author must add the inclusion and exclusion criteria in the study.

Response: A sub-section on inclusion/exclusion criteria is included in the revised version.

3. The research work Awareness of diagnosis, treatment plan and prognosis among patients attending public hospitals and health centers in Addis Ababa, Ethiopia is well conducted among the patients, the sample size is to be increased, and further work may be carried out.

Response: We agree with this suggestion and included it in our recommendations section

4. The authors also add the limitations of the study.

Response: Study limitations are described in the last paragraph of the discussion section

Reviewer #2: 

General comment: this study examines an important question about patients perception and understanding of care received. Inconsistencies in data and need of further clarity on methodology need to be examined.

Response: We have revised the manuscript to improve clarity and ensure consistency. 

Specifics:

Abstract: “Patients with good experience at their clinical assessment had significantly lower awareness of their prognosis"

Conclusion: “Patients with good experience at their clinical assessment had significantly higher awareness of their prognosis.”

 both statements are in contradiction, also "(OR=0.25; 95% CI: 0.08, 0.81)” in the abstract does not correspond to the data in the table

Response: We have revised the text in the conclusion section and verified the data presented in the abstract against table 3. 

Did the authors collect data on gender of health care providers? Was there a difference in patient knowledge of diagnosis, treatment plan or prognosis depending on provider gender? If provider gender was not examined would perhaps rephrase: “Patient care is shifting globally towards patient-centered approach [1, 2].”

Response: Data on gender of the provider was not collected. We have revised the text accordingly. 

“Questionnaire (PEQ) (25), Patient Perception of Patient-Centeredness (PPPC) scale (26), Perception on Quality of Health Service, General Patient Satisfaction Scale (27) and Items on Patients Awareness on their Diagnosis, treatment and Prognosis (28).”

was this translated or did the interviewer use the English version and each translate themselves? Were these instruments validated in the translated language? What language was used? Please provide a copy of the English and translated version as an appendix. Some validation of the instruments would be useful for this and future publications. This will also help readers to better understand how variables such as "patient centeredness" or "prognosis" or the various satisfaction/good care variables were defined.

Response: The English version of the tool was translated to Amharic by expert translators. The translated version was reviewed by experts and was the pre-tested before use. We have provided a copy of both the English and Amharic versions of the questionnaire.

Also authors could provide a reference indicating that documenting prognosis separate from the diagnosis and treatment plan is a requirement in medical notes. Is this now a requirement in Ethiopia, WHO, etc for example? Often prognosis is implicit in the diagnosis and treatment plan in medical notes.

Response: Yes, information about prognosis is also implicit in Ethiopia. We asked the patient if he or she knows whether his/her diagnosis is curable or not. For most of the information, we asked the treating physician about prognosis. Overall, prognostic information was largely inferred from the diagnosis. 

“We used logistic regression models to assess factors associated with awareness of diagnosis, treatment plan and prognosis”

What variables were included in these models? Has were interactions assessed and managed? It is not clear in the results section which results were bivariate, which were from multivariate models and or variables were included in the models.

Response: All the variables in table 3 were included in each logistic regression model (awareness of diagnosis, treatment and prognosis). We presented only adjusted ORs. We presented descriptive stats (bivariate) in table 2. In the revised version, we have indicated the variables and included in the models and that the ORs are adjusted.

Gynecologic patients had significantly lower awareness about their diagnosis and treatment plan, compared to who? Why is this different from females in general?

Response: This is compared to the reference category (patients in general outpatient department). We have controlled in the effect of gender in the model. But the reason behind this needs further study.

How could patients understand prognosis if they don’t know the diagnosis or treatment plan?

Response: The question we asked about prognosis is generic (whether the condition is treatable or not) and doesn’t need specific knowledge of diagnosis or details of treatment plan.

Also in the introduction would indicate the location of the references like in the discussion.

Response: We used the same reference style in both the introduction and discussion section. 

% in the text of the result would be better in a table.

Response: We have reduced some of the %s indicated in the text.

Data from the tables to keep tract

Awareness of Diagnosis Awareness of Treatment Awareness of Prognosis

Female

179(63.3) 163(54.2) 244(72)

1.40(0.91,2.15) 0.129 1.20(0.8,1.81) 0.374 1.17(0.73,1.86)0.511

Gynecology

31(48.4) 29(43.9) 64(77.1)

0.29(0.12,0.73) 0.009 0.30(0.12,0.77) 0.013 0.92(0.34,2.52) 0.879

General satisfaction (Ref: Not satisfied)

131(58.5) 116(48.5) 166(61.5)

0.68(0.43,1.05) 0.082 0.75(0.49,1.14) 0.179 0.75(0.47,1.19) 0.222

Response: These data from Table 2 and 3 go together. However, to keep the tables simpler, we put them in two tables.

6. PLOS authors have the option to publish the peer review history of their article (what does this mean?). If published, this will include your full peer review and any attached files.

Do you want your identity to be public for this peer review? For information about this choice, including consent withdrawal, please see our Privacy Policy.

Reviewer #1: No

Reviewer #2: No

---

## [Decision Letter · Decision Letter 1]

13 Dec 2021

PONE-D-20-36374R1Awareness of diagnosis, treatment plan and prognosis among patients attending public hospitals and health centers in Addis Ababa, Ethiopia.PLOS ONE

Dear Dr. Teklu,

Thank you for submitting your manuscript to PLOS ONE. After careful consideration, we feel that it has merit but does not fully meet PLOS ONE’s publication criteria as it currently stands. Therefore, we invite you to submit a revised version of the manuscript that addresses the points raised during the review process.

Dear Author,  This could be an important descriptive study. Kindly upload the questionnaire used. Prognosis assessment is important. The range of diagnosis is different from references cited. This is why this study may be important, but the authors should provide a detailed description of what they mean. Finally, statistics need to be addressed or regression model(s) removed and only publish bivariate analysis as a descriptive study. 

Also, address the comments by the reviewer. Please submit your revised manuscript by Jan 27 2022 11:59PM. If you will need more time than this to complete your revisions, please reply to this message or contact the journal office at plosone@plos.org. Please include the following items when submitting your revised manuscript:A rebuttal letter that responds to each point raised by the academic editor and reviewer(s). You should upload this letter as a separate file labeled 'Response to Reviewers'.A marked-up copy of your manuscript that highlights changes made to the original version. You should upload this as a separate file labeled 'Revised Manuscript with Track Changes'.An unmarked version of your revised paper without tracked changes. You should upload this as a separate file labeled 'Manuscript'.If applicable, we recommend that you deposit your laboratory protocols in protocols.io to enhance the reproducibility of your results. Protocols.io assigns your protocol its own identifier (DOI) so that it can be cited independently in the future. For instructions see: https://journals.plos.org/plosone/s/submission-guidelines#loc-laboratory-protocols. Additionally, PLOS ONE offers an option for publishing peer-reviewed Lab Protocol articles, which describe protocols hosted on protocols.io. Read more information on sharing protocols at https://plos.org/protocols?utm_medium=editorial-email&utm_source=authorletters&utm_campaign=protocols.

We look forward to receiving your revised manuscript.

Kind regards,

Kingston Rajiah

Academic Editor

PLOS ONE

Journal Requirements:

Reviewers' comments:

Reviewer's Responses to Questions

**Comments to the Author**

1. If the authors have adequately addressed your comments raised in a previous round of review and you feel that this manuscript is now acceptable for publication, you may indicate that here to bypass the “Comments to the Author” section, enter your conflict of interest statement in the “Confidential to Editor” section, and submit your "Accept" recommendation.

Reviewer #1: All comments have been addressed

Reviewer #2: (No Response)

2. Is the manuscript technically sound, and do the data support the conclusions?

Reviewer #1: Yes

Reviewer #2: Partly

3. Has the statistical analysis been performed appropriately and rigorously? 

Reviewer #1: Yes

Reviewer #2: No

4. Have the authors made all data underlying the findings in their manuscript fully available?

Reviewer #1: Yes

Reviewer #2: No

5. Is the manuscript presented in an intelligible fashion and written in standard English?

Reviewer #1: Yes

Reviewer #2: No

6. Review Comments to the Author

Reviewer #1: The authors have addressed all the queries raised by the reviewers and the article maybe accepted for publication.

Reviewer #2: Thank you to the authors for addressing issues

However, further clarifications would help

1-Decription of prognosis assessment in the African setting is important. More information how this was done would be helpful. For example, in the method section specifically described what questions were used and how was prognosis knowledge defined. Specifically, was it based on threat to life in short or long term, or possibility of disease chronicity? How were diagnosis in Ethiopian same or different to conditions in the reference's articles: cancer, schizophrenia in Western medicine setting? Explicit description of who with what diagnosis met prognosis understanding criteria would help.

2-The multiple regression analysis could use further clarity and description in the method section and table 3 (the title does not state that this is a multiple regression. Also it is not clear what are separate models, what are controlling variables, also interactions assessment is not addressed, the issue of multiple analysis alpha level adjustment mentioned by the editor also needs attention).

7. PLOS authors have the option to publish the peer review history of their article (what does this mean?). If published, this will include your full peer review and any attached files.

Reviewer #1: No

Reviewer #2: No

---

## [Author Response · Author response to Decision Letter 1]

6 Jun 2022

Responses to reviewers’ comments 

1-Decription of prognosis assessment in the African setting is important. More information how this was done would be helpful. For example, in the method section specifically described what questions were used and how was prognosis knowledge defined. Specifically, was it based on threat to life in short or long term, or possibility of disease chronicity? How were diagnosis in Ethiopian same or different to conditions in the reference's articles: cancer, schizophrenia in Western medicine setting? Explicit description of who with what diagnosis met prognosis understanding criteria would help.

Response: 

Prognosis in this study was determined based on threat to life in short or long term and need for continued treatment (chronicity). Patients were asked if their doctors have told them about possibility for cure (if they were told whether the treatment will cure them or if they needed to take it for life), as well as life expectancy life expectancy (threat to life). What the patients reported was compared with the patient record. Any patient with any condition who has been told that he will be cured after the treatment, any patient who has a knowledge of the chronicity of the disease and any patient who knows the estimated number of months/years s/he is left with, is categorized as “know their prognosis” when verified by the document and the panel of physicians. A revision is included in the manuscript. 

2-The multiple regression analysis could use further clarity and description in the method section and table 3 (the title does not state that this is a multiple regression. Also, it is not clear what are separate models, what are controlling variables, also interactions assessment is not addressed, the issue of multiple analysis alpha level adjustment mentioned by the editor also needs attention).

Response: We have refined the data analysis section based on your comments. The title of table 3 now indicates that the outputs are from multiple logistic regression. By separate models, we mean we ran three models- one for awareness of diagnosis as an outcome, a second for awareness of treatment, and a third for awareness of prognosis as outcome. As the multiple logistic regression analysis was exploratory, the OR for an independent variable were controlled for all other independent variables in the model. We have checked interactions among key variables in model 3. But we didn’t get any significant interaction. We appreciate the comment on alpha level adjustment. However, we think this would increase type II error.

---

## [Editor Report · Decision Letter 2]

10 Jun 2022

Awareness of diagnosis, treatment plan and prognosis among patients attending public hospitals and health centers in Addis Ababa, Ethiopia.

PONE-D-20-36374R2

Dear Dr. Teklu,

We’re pleased to inform you that your manuscript has been judged scientifically suitable for publication and will be formally accepted for publication once it meets all outstanding technical requirements.

Kind regards,

Kingston Rajiah

Academic Editor

PLOS ONE
---

## [Editor Report · Acceptance letter]

14 Jun 2022

PONE-D-20-36374R2 

Awareness of diagnosis, treatment plan and prognosis among patients attending public hospitals and health centers in Addis Ababa, Ethiopia. 

Dear Dr. Teklu:

I'm pleased to inform you that your manuscript has been deemed suitable for publication in PLOS ONE. Congratulations! Your manuscript is now with our production department. 

Kind regards, 

on behalf of

Associate Professor Kingston Rajiah 

Academic Editor

PLOS ONE